# Estimating Fractional Vegetation Cover of Row Crops from High Spatial Resolution Image

**Xu Ma** [1,*], **Lei Lu** [2], **Jianli Ding** [3], **Fei Zhang** [3] and **Baozhong He** [3]

1 Key Laboratory of Oasis Ecology of Ministry of Education, Postdoctoral Mobile Station, College of Resource and Environment Sciences, Xingjiang University, Urumqi 830064, China
2 College of Earth and Environmental Sciences, Lanzhou University, Lanzhou 730000, China; lulei@lzu.edu.cn
3 Key Laboratory of Oasis Ecology of Ministry of Education, College of Resource and Environment Sciences, Xingjiang University, Urumqi 830064, China; dingjl@xju.edu.cn (J.D.); zhangfei3s@xju.edu.cn (F.Z.); hbz108@163.com (B.H.)
* Correspondence: maxu2020@xju.edu.cn; Tel.: +86-13088706120

**Abstract:** With high spatial resolution remote sensing images being increasingly used in precision agriculture, more details of the row structure of row crops are captured in the corresponding images. This phenomenon is a challenge for the estimation of the fractional vegetation cover (FVC) of row crops. Previous studies have found that there is an overestimation of FVC for the early growth stage of vegetation in the current algorithms. When the row crops are a form in the early stage of vegetation, their FVC may also have overestimation. Therefore, developing an algorithm to address this problem is necessary. This study used World-View 3 images as data sources and attempted to use the canopy reflectance model of row crops, coupling backward propagation neural networks (BPNNs) to estimate the FVC of row crops. Compared to the prevailing algorithms, i.e., empirical method, spectral mixture analysis, and continuous crop model coupling BPNNs, the results showed that the calculated accuracy of the canopy reflectance model of row crops coupling with BPNNs is the highest performing (RMSE = 0.0305). Moreover, when the structure is obvious, we found that the FVC of row crops was about 0.5–0.6, and the relationship between estimated FVC of row crops and NDVI presented a strong exponential relationship. The results reinforced the conclusion that the canopy reflectance model of row crops coupled with BPNNs is more suitable for estimating the FVC of row crops in high-resolution images.

**Keywords:** high spatial resolution image; fractional vegetation cover; row crops; neural networks





## 1. Introduction

Fractional vegetation cover (FVC) is a canopy architecture parameter that represents the fraction of the land surface covered by green foliage in the two-dimensional plane [1]. FVC is a very important parameter for describing the vegetation cover on the Earth's surface, and also is one of the important indicators of ecosystem change [2]. Vegetation usually includes crops, forests, grasslands, and shrubs. Different from other vegetation, the crops are artificially planted. Its FVC is closely related to the growth status of the crop, which can further estimate the yield of food. Therefore, crop FVC is very important in agriculture. With the development of information technology, precision agriculture was proposed. Precision agriculture requires remote sensing as support [3,4]. In the estimation of FVC during the crop growing season, high spatial resolution remote sensing data is a good choice for precision agriculture [5]. When crops are being sowed in the row mode, also known as row crops, it consists of row closures (vegetation component) and between-row (bare soil component) in the early growth stage [6]. The estimation of the FVC of row crops is more complicated than that of continuous crops (canopy state at the later stage of crop growth) in the high spatial resolution remote sensing data. Therefore, developing the estimated methods of the FVC of row crops are urgently needed in remote sensing.

To estimate FVC from remote sensing images, the prevailing algorithms include the (1) empirical method, (2) spectral mixture analysis (SMA), and (3) physical model-based method [7–9]. In the empirical method, the regression relationship (this relationship can also be understood as causality) between the vegetation index (VI) and FVC is developed. Then, this relationship is used to estimate the FVC [8]. This method has high calculation accuracy, which is suitable for estimating FVC based on regional remote sensing data and is further used as reference data to validate other methods [7,10]. However, the empirical method lacks a physical mechanism and requires a large number of samples to ensure the accuracy of the regression relationship [9]. Therefore, this method is not suitable for the estimation of FVC over a large area. Moreover, when an image contains heterogeneous canopies, the accuracy of the empirical method is limited [11,12]. Row crops are typical heterogeneous canopies. Therefore, the empirical method is not a good choice for developing a method to estimate the FVC of row crops.

SMA is also called a pixel unmixing model, and it overcomes the limitations of the empirical method for estimating FVC over a large area [13]. SMA was developed based on early vegetation micrometeorological modeling [12,14,15], and extended some similar algorithms, e.g., the pixel dichotomy model [1,16]. In SMA, the endmember (the decomposed "basic component" of the pixel) probability in each pixel is constructed, and then is combined with spectral linear equation and classification algorithm to calculate FVC [17]. SMA is a semi-empirical algorithm, and considers spectral mixing relationships in the modeling, which has a certain physical mechanism [18]. According to solutions of the linear equations [17], SMA is divided into unconstrained inversion and constrained inversion [19,20]. Different from unconstrained inversion, constrained inversion considers constrained conditions (the sum of all endelements is 1 and all endelements are greater than 0) [21]. Therefore, this method will not have outliers (FVC appears greater than 1 or less than 0) when estimating FVC, and is rarely used in high spatial resolution images [20,22]. The reason is that the heterogeneity of ground objects in the images becomes more obvious with the increase of spatial resolution, which becomes a challenge for the selection of endmembers in SMA [23]. Therefore, whether SAM is a good method to estimate the FVC of row crops needs further study.

Different from the empirical method and SMA, the physical model-based method relies on radiative transfer models. The direct estimation of radiative transfer models is very difficult due to the complexity of the models. Usually, neural networks (NNs) and lookup table methods are the two typical alternative methods for the indirect estimation of physical models [7,24]. NNs are the most commonly used method [7]. In NNs, the radiative transfer models produce a learning dataset to "train" and "test" NNs. Currently, many FVC products from remote sensing have been developed based on this method, e.g., POLDER products, CYCLOPES products, GEOV1 products, and GLASS-FVC products [25–27]. In previous studies, since the spatial resolution of remote sensing data was very low, the spatial resolution of the corresponding FVC products was also very low. Therefore, the radiative transfer models usually use continuous crop models to approximately calculate the FVC of row crops [7]. However, with continuous improvement on the spatial resolution of the sensors, more details of the row structure of the row crops are captured in the corresponding images. Continuous crop models were continued to use to estimate the FVC of row crops, which will cause overestimation [28]. Therefore, it is necessary to introduce a canopy reflectance model of row crops to develop a new estimated method.

The objective of this study will develop a general and reliable FVC estimation algorithm for row crops in the high spatial resolution images, and further address the overestimation in the FVC of row crops. To achieve this goal, (1) we use NNs and canopy reflectance model of row crops to participate in estimating FVC of row crops, then compare the empirical method, SMA, and continuous crop model coupling BPNNs in the high spatial resolution images. (2) By analyzing the accuracy for differences among prevailing algorithms of FVC, we can find an optimal algorithm to estimate the FVC of row crops. Through the above two steps, this study can demonstrate the advantages of the canopy

reflectance model of row crops in estimating the FVC of row crops, and find a suitable estimated algorithm for the FVC of row crops.

## 2. Methods

### 2.1. Empirical Method

In the empirical method, the regression equation between the normalized difference vegetation index (NDVI) and FVC is developed based on the training data [29–33], namely:

$$FVC = P(0°) = a\text{NDVI} + b \tag{1}$$

Here, $P(0°)$ is the gap probabilities in the vertical viewing direction, i.e., FVC. $a$ and $b$ are empirical coefficients, which can be fitted based on field measurements. According to the measurement results of this study (see Figure 3c below), $a = 0.755$, $b = -0.079$.

### 2.2. Spectral Mixture Analysis

With the assumption that a pixel is the linearly weighted sum of each component endmember, the SMA uses the spectral linear equation (i.e., Equation (2)) and pixel classification algorithm (we used orthogonal subspace projection, i.e., OSP, see Section 2.2.1) [34] to calculate FVC [35]. The expression of the spectral linear equation is

$$R_k = \sum_{i=0}^{M} r_{i,k} f_i + e_k \tag{2}$$

Here, $R_k$ is the reflectance of the $k$th band, $i$ is the number of endelements, $f_i$ is the abundance (area fraction) of the $i$-th spectral endmember in the pixel, $r_{i,k}$ is the reflectance of the $i$-th endmember in the $k$-th band, and $e_k$ is the residual term of the spectral linear equation, i.e., the calculated deviation. Equation (2) is rewritten as matrix-vector form:

$$\mathbf{R} = \mathbf{r}\mathbf{F} + \mathbf{E} \tag{3}$$

According to the difference of solution methods used for spectral linear equation, i.e., Equations (2) or (3), SMA is divided into unconstrained SMA(U-SMA) [36] and constrained SMA (C-SMA) [17]. The detailed calculation is shown in the following subsections.

#### 2.2.1. Unconstrained SMA

$\mathbf{F}$ in Equation (3) represents the vector of the abundance (area fraction), one of which is the abundance (area fraction) of vegetation components, i.e., FVC. Therefore, we decompose $\mathbf{F}$ in Equation (3), and the equation is

$$\mathbf{R} = \mathbf{r}_0 f_0 + \mathbf{r}_{other}\mathbf{F}_{other} + \mathbf{E} \tag{4}$$

Here, $f_0$ is FVC, $\mathbf{r}_0$ is vector corresponding to $f_0$, $\mathbf{r}_{other}$ is $\mathbf{r}$ excluding $\mathbf{r}_0$, and $\mathbf{F}_{other}$ is $\mathbf{F}$ excluding $f_0$. To remove $\mathbf{r}_{other}\mathbf{F}_{other}$ in Equation (4), we introduce orthogonal projection. i.e., $\mathbf{P} = \mathbf{I} - \mathbf{r}_{other}\mathbf{r}_{other}^+ = \mathbf{I} - \mathbf{r}_{other}(\mathbf{r}_{other}^T\mathbf{r}_{other})^{-1}\mathbf{r}_{other}^T$. Here, $\mathbf{P}$ is an operator symbol of orthogonal projection, and $\mathbf{r}_{other}^+$ is the pseudoinverse of $\mathbf{r}_{other}$. Then, Equation (4) becomes:

$$\mathbf{P}(\mathbf{R}) = \mathbf{P}(\mathbf{r}_0 f_0) + \mathbf{P}(\mathbf{E}) \tag{5}$$

We then use the signal-to-noise ratio (SNR) algorithm [34] to calculate maximum eigenvalue. Then, Equation (5) can be solved and the FVC ($f_0$ in Equation (5)) can be calculated.

2.2.2. Constrained SMA

Compared with U-SMA, C-SMA includes two constraints: (1) the sum of components equals one, and (2) the component value is greater than zero, which makes C-SMA closer to the real situation. The constraint is expressed as the mathematical form:

$$
\min_{f \in \Delta} \left\{ (\mathbf{R} - \mathbf{Fr})^{\mathrm{T}} (\mathbf{R} - \mathbf{Fr}) \right\}
$$
$$
s.t. \Delta = \left\{ f \middle| \sum_{j=1}^{n} f_j = 1 \right\}
$$
$$
s.t. \Delta = \{ f > 0 \}
$$

(6)

In this study, we used the constrained least squares to address the first constraint. The second constraint is an issue of extreme value in mathematics, and we introduced the Lagrange multiplier method to calculate the eigenvalue vector in Equation (6). After that, we used an iterative algorithm to solve Equation (6), and Kuhn-Tucker conditions are set to convergence conditions.

2.2.3. Number of Endmembers

The number of endelements in the SMA influences the estimated accuracy of FVC. In [21], the SMA of five endmembers (SMA-5) has high accuracy to simulate FVC in the heterogeneous canopy. The canopy of row crops is a typical heterogeneous canopy in agriculture. Therefore, we chose SMA-5 to participate in this comparison. Meanwhile, considering that the row crops were mainly composed of vegetation and background, we also chose the SMA of the two endmembers (SMA-2). Therefore, the SMA includes four cases in this study, i.e., U-SMA-2, U -SMA-5, C-SMA-2, and C-SMA-5.

*2.3. Physical Model-Based Method*

The basis of the physical model-based method is the radiative transfer model. Therefore, we chose the canopy reflectance model of row crops to address the simulated overestimation of FVC of row crops. In this study, a modified four-stream (MFS) radiative transfer model is used, which is developed based on the four-stream radiative transfer theory [6]. In the MFS model, the horizontal radiative transfer caused by the change of row structure is considered; hence, the MFS model is most suitable for the canopy of row crops. Based on the light transmittance, the MFS model establishes the relationship between the light signal and the gap in the row crops. Then, FVC is calculated based on gap fraction from nadir observations. In addition, to show the accuracy of the continuous crop reflectance model to estimate the FVC of row crops in the high spatial resolution images, we had chosen the PROSAIL model [37] to participate in the comparison. The PROSAIL model assumes that the horizontal direction of the canopy is infinitely [38]. Therefore, the PROSAIL model is suitable for estimating the FVC of the continuous crop. The input parameters of the radiative transfer models (MFS and PROSAIL models) are generally a lot, and some input parameters are difficult to be measured, which makes them difficult to solve, and further to acquire FVC [7]. Therefore, in this study, we introduced backward propagation NNs (BPNNs) to address this problem. Use of BPNNs is one of the NNS methods, their training time is shorter, and robustness is strong in the estimation. Therefore, this method is widely used to find the optimal solution for the radiative transfer model in Earth sciences [39]. The calculation steps are as follows:

2.3.1. Generating the Learning Dataset

Since BPNNs need the learning dataset (a "one-to-many" data table composed of FVC and each band of the spectrum, the band of the spectrum can be seen in Table 2) to participate in "training" and "testing", we used the MFS model to generate a learning dataset. Since the PROSPECT-5 model (leaf reflectance model) [40,41] provided leaf hemispherical reflectance and transmittance to the MFS model, we coupled the PROSPECT-5 model to the

MFS model (canopy reflectance model). In the generation of the learning dataset of the MFS model, the range of the leaf biochemical parameters involved in the PROSPECT-5 model refers to Leaf Optical Properties Experiment 93 (Lopex 1993) [42], and these parameters are: leaf chlorophyll content ($C_{ab}$), carotenoid content ($C_{ar}$), brown pigment content ($C_{brown}$), equivalent water thickness ($C_w$), leaf mass per unit leaf area ($C_m$), and structure coefficient ($N$). In addition, for other canopy parameters in the MFS model, we chose leaf area index ($L$), average leaf inclined angle ($\theta_l$), row width ($A_1$), and distance between rows ($A_2$) to participate in the generation of learning dataset. The PROSAIL model is also composed of the PROSPECT-5 model and SAIL model (canopy reflectance model) [38]; hence, we also used a similar method to generate a learning dataset of the PROSAIL model. Table 1 shows an example of the input parameters used to generate a learning dataset.

**Table 1.** The input parameters for the MFS model used to generate the learning dataset.

| MFS Model | | | | PROSAIL Model | | | |
|---|---|---|---|---|---|---|---|
| Parameter | Unit | Value Range | Step | Parameter | Unit | Value Range | Step |
| $C_{ab}$ | $\mu g \cdot cm^{-2}$ | 30–60 | 10 | $C_{ab}$ | $\mu g \cdot cm^{-2}$ | 30–60 | 10 |
| $C_{ar}$ | $\mu g \cdot cm^{-2}$ | 4–14 | 2 | $C_{ar}$ | $\mu g \cdot cm^{-2}$ | 4–14 | 2 |
| $N$ | - | 1.2–1.8 | 0.2 | $C_W$ | cm | 0.005–0.0015 | 0.005 |
| $\theta_l$ | ° | 30–70 | 10 | $C_m$ | $g \cdot cm^{-2}$ | 0.003–0.005 | 0.001 |
| $L$ | $m^2 \cdot m^{-2}$ | 0–6 | 0.5 | $N$ | - | 1.2–1.8 | 0.2 |
| $\theta_s$ | ° | 20–45 | 5 | $\theta_l$ | ° | 30–70 | 10 |
| $A_1$ | cm | 0–20 | 10 | LAI | $m^2 \cdot m^{-2}$ | 0–6 | 0.5 |
| $A_2$ | cm | 0–20 | 10 | $\theta_s$ | ° | 20–45 | 5 |

Here some parameters in MFS model are fixed, they are: $C_w$ = 0.014, $C_m$ = 0.004, $C_{brown}$ = 0, h = 1, hotspot = 0.1. Similarly, some parameters are also fixed in the PROSAIL model, they are: $C_{brown}$ = 0, hotspot = 0.1.

In the learning dataset, the MFS model and the PROSAIL model generate 168,480 cases, respectively. After the learning dataset is generated, we randomly divided the learning dataset into three parts: 60% of the learning dataset is used to "train" BPNNs, and this learning dataset is also called the training dataset; 20% of the learning dataset is used to validate the network, and this learning dataset is also called the validating dataset. The remaining learning dataset was used to "test" the network, and this learning dataset is also called the testing dataset.

### 2.3.2. Training Network

A BPNN includes an input layer, hidden layer, and output layer. In this step, we used the "training dataset" and "validation dataset" to train the network. The calculation of BPNNs is divided into two steps: forward propagation and backward error propagation. Based on the above two steps, the final parameters of the network can be determined; hence, the "test dataset" is used to produce the final reference data table.

In forward propagation, the activation function is used to construct a network. This study uses the Sigmoid function, and its expression is:

$$f(I) = \frac{1}{1 - e^I} \tag{7}$$

where $I$ is a neuron function, which is commonly expressed as a one-dimensional linear function, and its expression is $w_{ji}y_j + b_j$. $w_{ji}$ is the weight in the network, $y_j$ is the output value of the network, and $b_j$ is the bias of the network.

In backward error propagation, the momentum method (hyperparameters) is used to adjust the error to the minimum in the network. It improves the "learning efficiency" (to

find the optimal solution of the network at the shortest time, the "training" of less than ten cycles almost achieves the minimum deviation, see Figure 1b,f), and its expression is:

$$w_{ij}(p+1) := w_{ij}(p) + \alpha \frac{\partial F}{\partial w_{ij}} + \beta \left[ w_{ij}(p) - w_{ij}(p-1) \right] \tag{8}$$

$$b_j(p+1) := b_j(p) + \alpha \frac{\partial F}{\partial b_j} + \beta \left[ b_j(p) - b_j(p-1) \right] \tag{9}$$

where $:=$ is a mathematical symbol of "defined as", $\alpha$ is the learning efficiency, and its expression is $\alpha := \alpha / \lambda(i)$. This expression is an adaptive function which automatically adjusts step length to find the optimal solution in the training. $\beta$ is the damping coefficient (or momentum), which updates the previous gradient value to speed up the convergence. The range of the value is $0 < \beta < 1$. $P$ is the number of network cycles. $F$ is the cost function, which is used to control the error. The cost function used in this study is

$$F = \sum_{i=1}^{p} \sum_{j=1}^{l} \left( y_{ij} - \hat{y}_{ij} \right) \tag{10}$$

where $\hat{y}_{ij}$ is the estimated value when the BPNN is "trained". Here, the MFS model combined with the BPNN is referred to as the MFS + BPNN, and the PROSAIL model combined with a BPNN is referred to as the PROSAIL + BPNN. The number of nodes in the hidden layer has a large influence on the performance of BPNNs. To determine the number of nodes in the hidden layer, we analyzed the variation of deviation during the network cycles in the MFS + BPNN and PROSAIL + BPNN (Figure 1). When the number of nodes hidden layer is 6, the deviation of the MFS + BPNN is the smallest (blue line in Figure 1c or red box in Figure 1d). Therefore, the number of nodes in the hidden layer in the MFS + BPNN is set to 6. Similarly, the number of nodes in the hidden layer in the PROSAIL + BPNN is set to 14 (purple line in Figure 1g or red box in Figure 1h). The other parameters in BPNNs can be seen in Table 2.

**Table 2.** The input parameters of the BPNNs in the program.

| | Number of Network Cycles (P) | Number of Nodes in the Layers | | | Learning Efficiency ($\alpha$) | Damping Coefficient ($\beta$) |
| | | Input | Hidden | Output | | |
|---|---|---|---|---|---|---|
| value | 414 | 3 | 6 (MFS + BPNN) | 1 | | |
| specific contents | | 545 nm (r) 660 nm (r) 782 nm (r) | 14 (PROSAIL_BPNNs) | FVC | 0.01 | 0.1 |

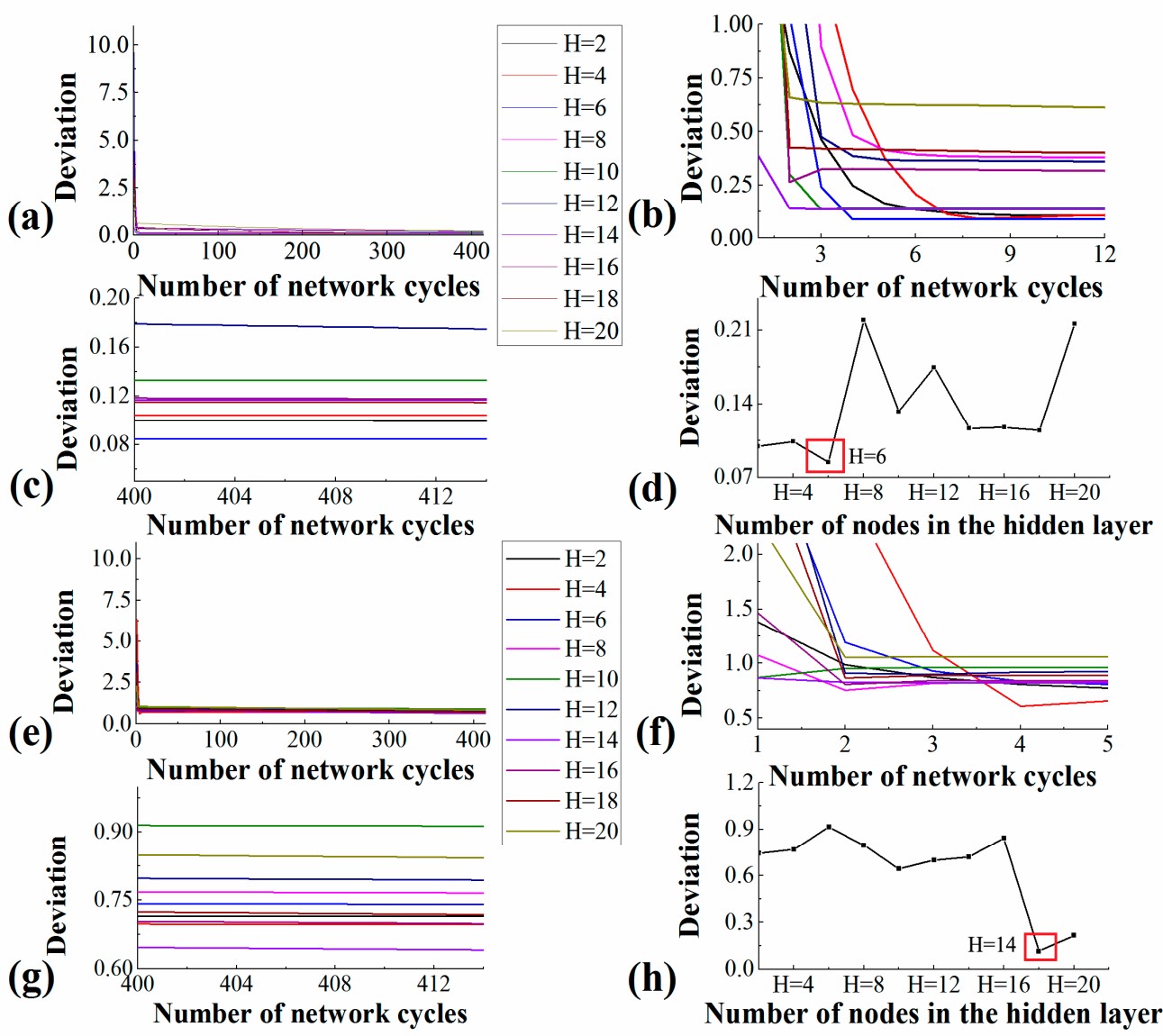

**Figure 1.** The variation of deviation during network cycles in the MFS + BPNN and PROSAIL + BPNN. (**a**) The variation of deviation during network cycle in the MFS + BPNN, (**b**) the most obvious position for deviation reduction in MFS + BPNN, (**c**) the deviation of the MFS + BPNN at the last few cycles, and (**d**) the deviation of the MFS + BPNN at the last cycle, (**e**) the variation of deviation during network cycle in MFS + BPNN, (**f**) the most obvious position for deviation reduction in the MFS + BPNN, (**g**) the deviation of the MFS + BPNN at the last few cycles, and (**h**) the deviation of the MFS + BPNN at the last cycle. Here, red boxes in Figure 1d,h are the number of nodes in the hidden layer when the deviation is minimum, and the symbol H is an abbreviation for the number of nodes in the hidden layer.

### 2.3.3. Retrieving FVC

When the network is trained, the "test dataset" is substituted into the network. Then, an "output test dataset" can be generated. We used the "output test dataset" to retrieve the FVC corresponding to the spectrum in the pixels. In this study, we used spectral angle as a retrieving algorithm.

$$\vartheta = \arccos\left(\frac{\mathbf{A} \cdot \mathbf{B}}{|\mathbf{A}||\mathbf{B}|}\right) \tag{11}$$

where **A** and **B** are the reflectance of the two spectra on the bands. In this study, the MFS + BPNN and the PROSAIL + BPNN used spectral angle to retrieve FVC, and their threshold was 0.05.

## 3. Experiment and Data Preparation

In this study, the in-situ measurements were collected in Zhongwei City, Ningxia Hui Autonomous Region, China (Figure 2a). The Yellow River passes through the region, and Zhongwei City is a typical area of irrigation agriculture. In Zhongwei City, the farmland on the south bank of the Yellow River is more concentrated. Therefore, this study chose farmland on the south bank of the Yellow River as a study area (Figure 2b). The main species of the study area include rice, corn, and wolfberry (strictly speaking, wolfberry is an economic forest planted in row mode). For the estimation issues of FVC of row crops, we measured the early growth status of the crop, and its measurement time was performed from 1–2 July 2016. In this time, the measured crops exhibited a significant row structure. To be consistent with the high spatial resolution (World-View 3 image in Figure 2c), the plot area was selected 2 m × 2 m. The specific correlation measurements are shown below:

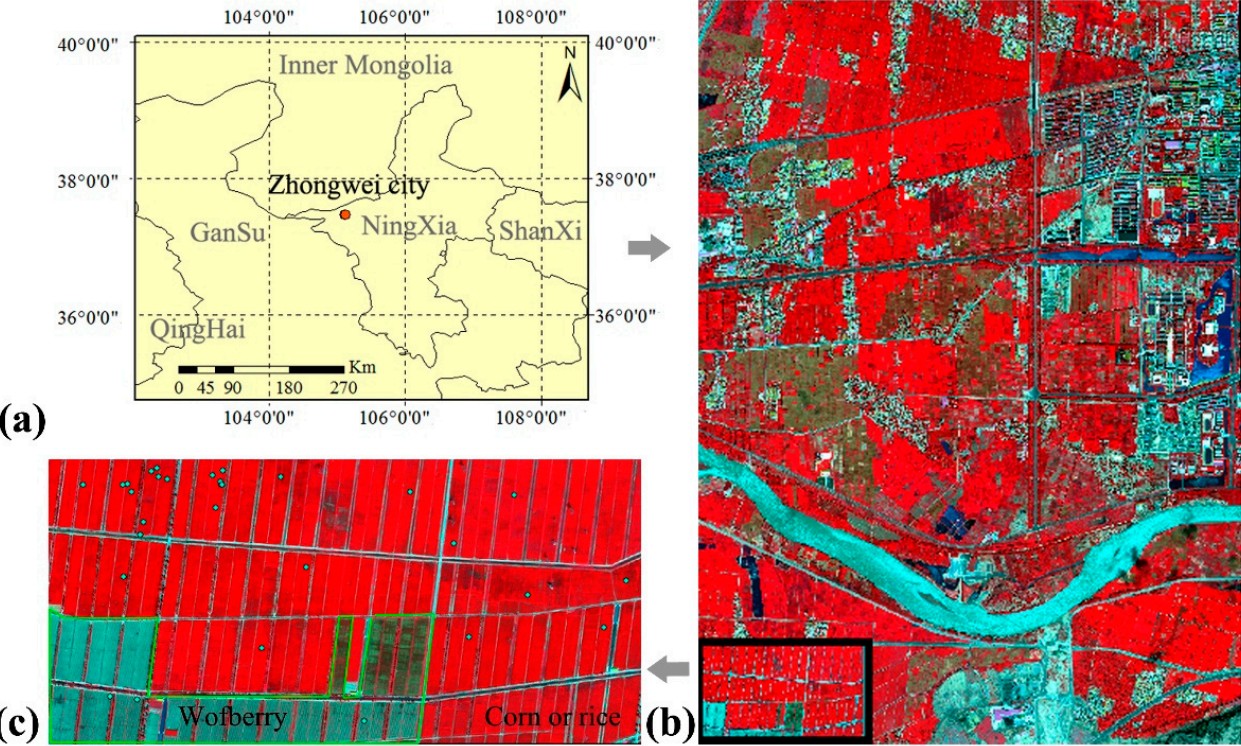

**Figure 2.** Map and false-color images of the study area. (**a**) Map of the study area, (**b**) World-View 3 image in Zhongwei city, and (**c**) World-View 3 image of farmland in Zhongwei city. The green points in (**c**) are the measuring sample, the blue area in (**c**) is the planting area of wofberry, and the red area in (**c**) is the mixed planting area of rice and corn.

### 3.1. Field Measurement Data

3.1.1. Endmember Spectrum and Normalized Vegetation Index

There are two ways to obtain the spectrum of endmembers. The first way is directly obtained from remote sensing images through pure pixel analysis [33], and the second way is obtained via ground measurement. Since pure pixel analysis is more convenient, we used it to extract endmember spectrum in the World-View 3 image. Figure 3a,b shows the endmember spectrum for U-SMA-2, U-SMA-5, C-SMA-2, and C-SMA-5. In NDVI measurement, we selected 30 samples sites involved in the remote sensing image and used a canopy analyzer (SperctroSense 2+ multi-channel canopy spectrum measurement system) to measure the normalization difference vegetation index (NDVI) (Figure 2c). Based on the measured NDVI and the measured FVC (Section 3.1.2), we established a regression equation to calculate the FVC of row crops (Figure 3c).

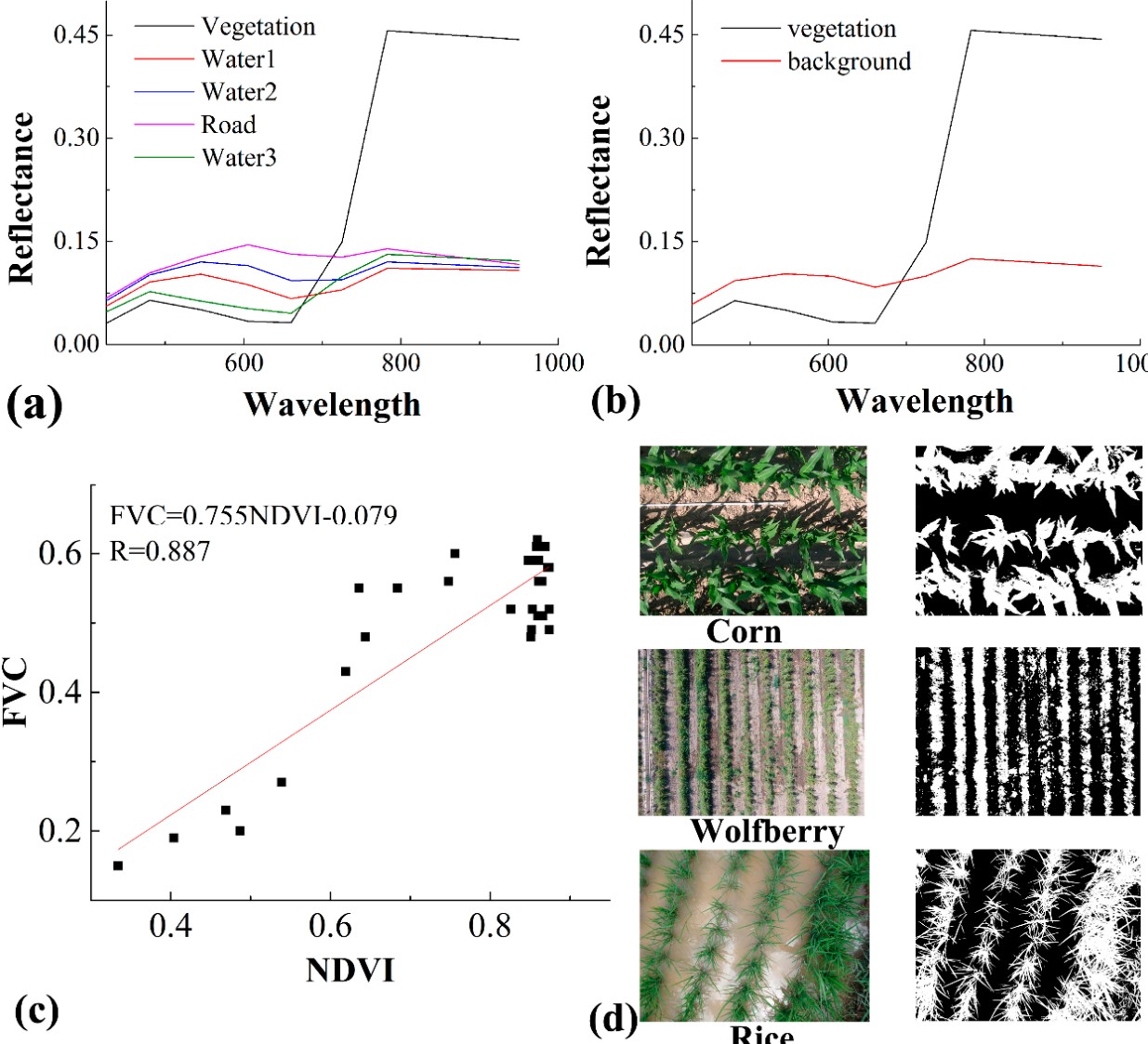

**Figure 3.** Endmembers, regression relationship, and image involved in digital photography. (**a**) The endmembers involved in U-SMA-2 and C-SMA-2, (**b**) the endmembers involved in U-SMA-5 and C-SMA-5, (**c**) the regression relationship between measured FVC and measured NDVI, and (**d**) the RGM images and binary images involved in digital cover photography.

### 3.1.2. Fractional Vegetation Cover

In FVC measurement, we selected 30 samples sites involved in the remote sensing image (Figure 2b). Digital cover photography (DCP, non-fisheye lens) was used to measure the FVC of row crops. According to the different types of crops, the measured distance between the camera lens and the top of the canopy was between 2 m, and the number of photos in each quadrat was not less than three to obtain the average value of FVC. In the processing photos taken by DCP, we used Lab's spatial conversion technology to convert the captured color photos into binary images. In the binary images, the pixel value of 0 represented the soil, and the pixel value of 255 represented the area covered by vegetation. We counted the number of pixels in the vegetation component to obtain FVC in the photos (Figure 3d).

### 3.2. Remote Sensing Data, Preprocessing, and Implementation of Estimated Algorithm of FVC

In this study, we used a high spatial resolution image (World-View 3) as remote sensing data. The related introduction of World-View 3 is shown in Table 3. In remote

sensing preprocessing, we used ENVI software to process the image. First, we performed a radiometric calibration on the image and converted the DN value into radiance. Then, based on ground control points, we performed a geometric correction in the images. To obtain the reflectance of ground objects, we used FLAASH (Fast Line-of-sight Atmospheric Analysis of Spectral Hypercubes) in the ENVI software to conduct an atmospheric correction on World-View 3 image [43]. Moreover, we used IDL computer language to write all the estimated algorithms of FVC involved in this study, i.e., empirical method, U-SMA-2, U-SMA-5, C-SMA-2, C-SMA-5, MFS + BPNN, and PROSAIL + BPNN. The version of IDL was 8.3 Microsoft Windows (Win32 x86_64 m64). To address images in ENVI software, we used the image processing module in the ENVI function library.

**Table 3.** The introduction of World-View 3.

| Satellite Height | Band | Acquisition Date and Time | Spatial Resolution |
| --- | --- | --- | --- |
| 617 km | Coastal (400 nm–450 nm) Blue (450 nm–510 nm) Green (510 nm–580 nm) Yellow (585 nm–625 nm) Red (630 nm–690 nm) Red–Edge(705 nm–74 nm) NIR–1 (770 nm–895 nm) NIR–2 (860 nm–1040 nm) | 2 July, 12:15:34 (Beijing time) | 2 m |

## 4. Results

Figure 4 shows the comparison of FVC of row crops estimated from different algorithms in World-View 3. The FVC values of row crops estimated by U-SMA-2, C-SMA-2, U-SMA-5, and C-SMA-5 (Figure 4c–f) are significantly higher than the FVC values of row crops estimated by the MFS + BPNN, PROSAIL + BPNN, and empirical method (Figure 4a,b,g). In Figure 4, vegetation planted in the red area is wolfberry, and its FVC is low. Vegetation planted in the khaki area (Figure 4a,b,g) or green area (Figure 4c–f) are rice or corn, and their FVC is high. In U-SMA-2, C-SMA-2, U-SMA-5, and C-SMA-5, most FVC values are estimated by U-SMA-2, C-SMA-2, U-SMA-5, and C-SMA-5 are high, and the values are located at 0.9 (red or yellow area in Figure 5c–f). The FVC values estimated by the MFS + BPNN, PROSAIL + BPNN, and empirical method are low, and the most values are located in 0.5–0.6 (red or yellow area in Figure 5a,b,g)). Compared with the SMA in Figure 5c–f, the scatter points in Figure 5a,b,g are more concentrated, which implies the FVC of row crops estimated by the MFS + BPNN, PROSAIL + BPNN, and empirical method are strongly related with NDVI. There is a significant exponential relationship between NDVI and FVC estimated by the MFS + BPNN and PROSAIL + BPNN (Figure 5a,b). In the exponential relationship between NDVI and FVC, the continuity of results estimated by the MFS + BPNN is better than that of the PROSAIL + BPNN. It is worth noting that these discontinuous points displayed in Figure 5b are shown as the steep change of grayscale, i.e., green point distribution in the yellow area in Figure 5b. The maximum FVC estimated by the PROSAIL + BPNN is greater than that of the MFS + BPNN (red circle in Figure 5b). In Figure 5g, the FVC of row crops estimated by the empirical method has linear relations with NDVI. In the U-SMA-2 and U-SMA-5, FVC has many values greater than 1 or less than 0, see the points outside the red dotted line in Figure 5c,e. The FVC of row crops estimated by C-SMA-2 and C-SMA-5 is between 0 and 1.

In Figure 6, the field measurement data is used to evaluate the performance of estimated algorithms of the FVC of row crops for World-View 3 data. The scatter point of the FVC estimated by MFS + BPNN and those calculated from the field photos (Figure 3d) have high consistency, i.e., R = 0.9763, RMSE = 0.0305 in Figure 6a. The overall performance using all the field measurement data presented satisfactory results, which further confirms the reasonability and reliability of the MFS + BPNN. In Figure 6b,g), the calculation accuracy of the PROSAIL + BPNN (R = 0.9166, RMSE = 0.0665) and empirical method (R = 0.9463,

RMSE = 0.0378) is lower than that of MFS + BPNN. Some scatter points are distributed in the upper left of the 1: 1 line, which implies that the PROSAIL + BPNN and empirical method have a slight overestimation when estimating FVC of row crops (Red circle in Figure 6b,g). The row crop' FVC estimated by U-SMA-2, C-SMA-2, U-SMA-5, and C-SMA-5 presents a significant overestimation, and its calculation accuracy is low (RMSE > 0.0792). In general, the estimated results of U-SAM-5 and C-SAM-5 (RMSE > 0.0792) are better than that of U-SMA-2 and C-SMA-2 (Their RMSE > 0.0755). In Figure 6c–f, the difference between RMSEs in U-SMA-2, C-SMA-2, U-SMA-5, and C-SMA-5 is small. Combined with Figure 5c–f, constraint conditions are considered, which avoids outliers. However, constraint conditions can be ignored in SMA for the improvement of estimation accuracy of the FVC of row crops in World-View 3.

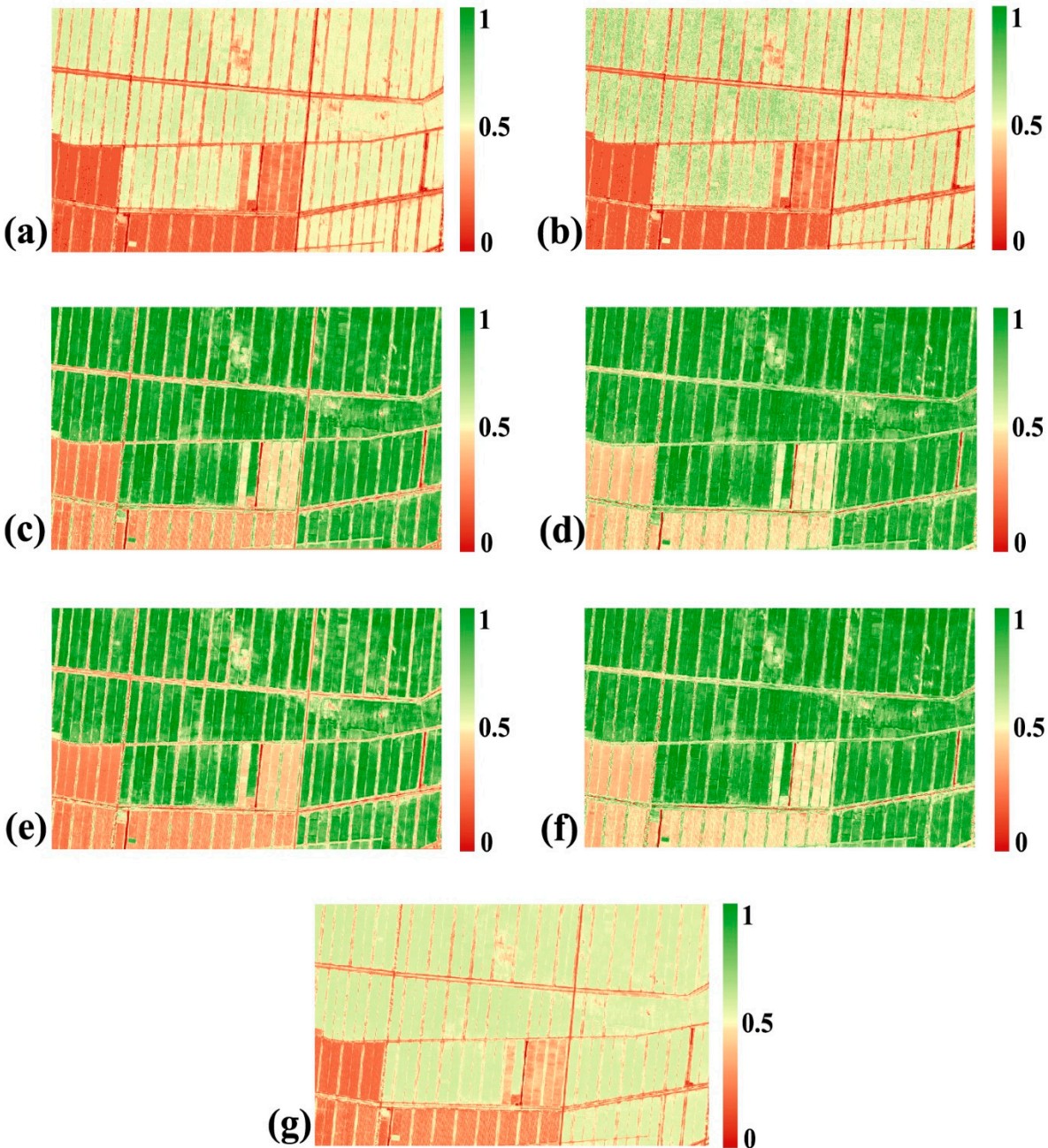

**Figure 4.** The images of FVC of row crops estimated from the different algorithms for the World-View 3. (**a**) MFS + BPNN, (**b**) PROSAIL + BPNN (**c**) U-SMA-2, (**d**) C-SMA-2, (**e**) U-SMA-5, (**f**) C-SMA-5, and (**g**) empirical method.

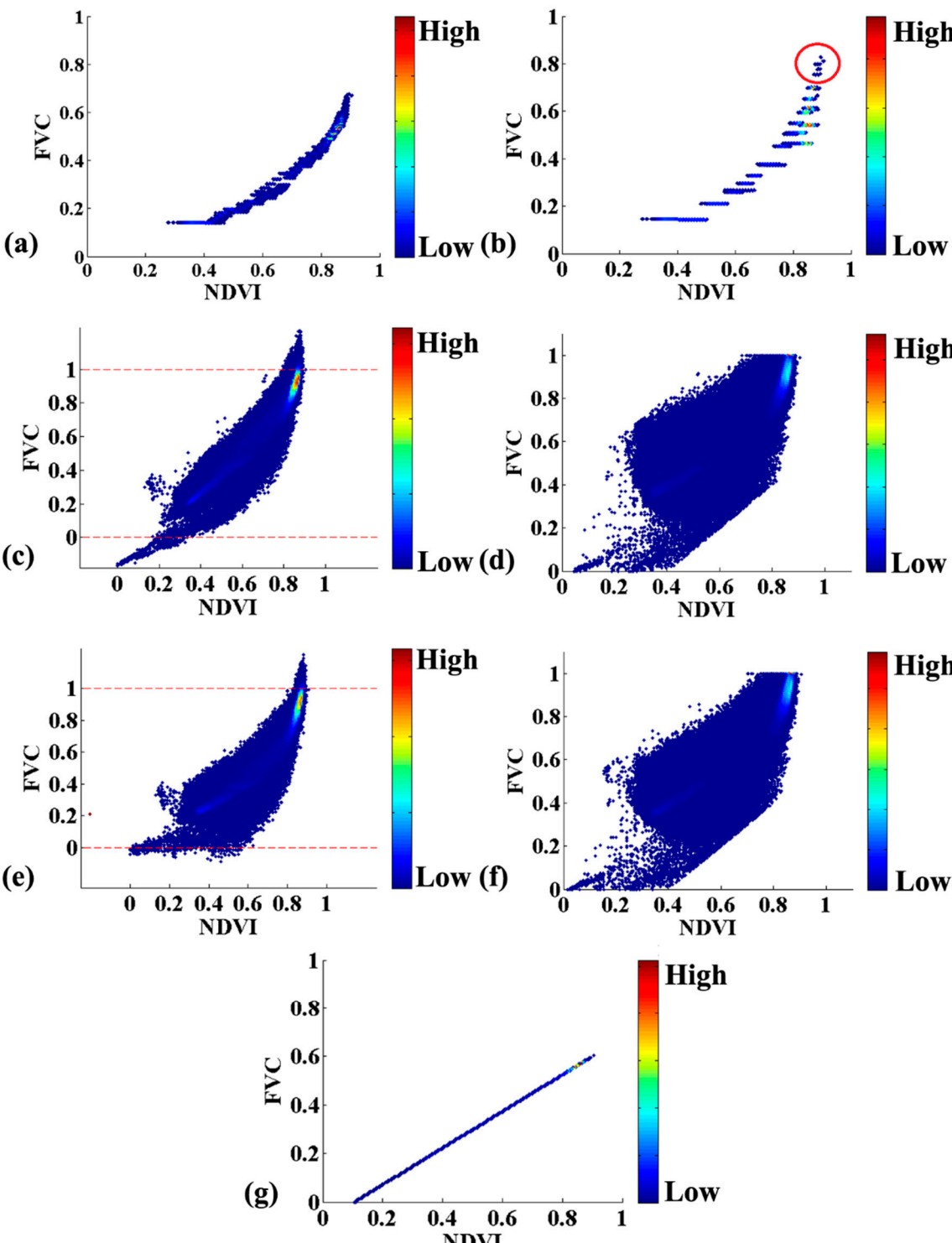

**Figure 5.** The density plots of the relationship between NDVI and FVC estimated from different algorithms in the World-View 3. (**a**) MFS + BPNN, (**b**) PROSAIL + BPNN, (**c**) U-SMA-2, (**d**) C-SMA-2, (**e**) U-SMA-5, (**f**) C-SMA-5, and (**g**) empirical method.

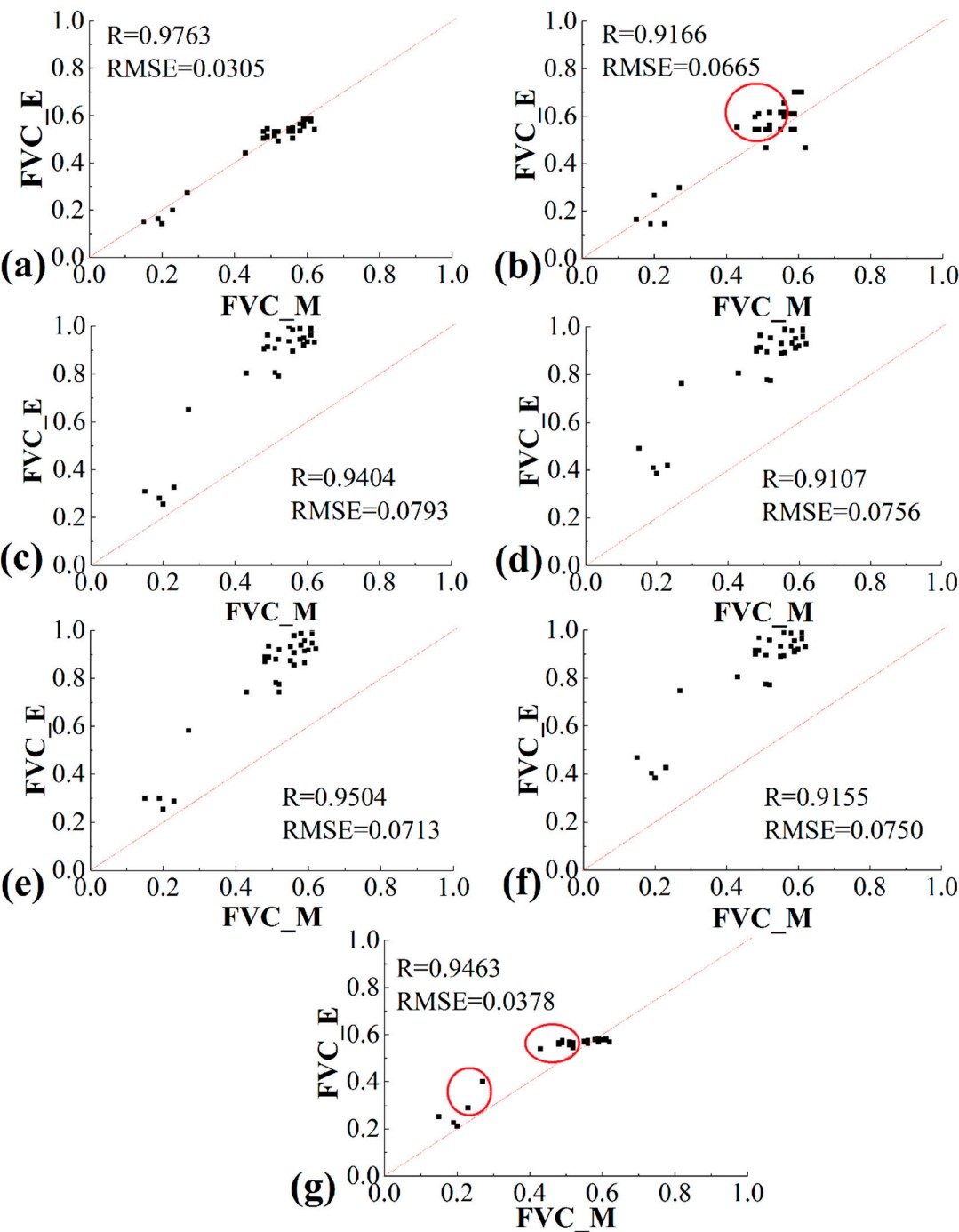

**Figure 6.** The Statistics of the FVC of row crops simulated by the different algorithms in the World-View 3 and field measurement data. (**a**) MFS + BPNN, (**b**) PROSAIL + BPNN (**c**) U-SMA-2, (**d**) C-SMA-2, (**e**) U-SMA-5, (**f**) C-SMA-5, and (**g**) empirical method. Here FVC_E is estimated FVC, FVC_M is measured FVC, and the red circle represents an overestimated area.

## 5. Discussion

This study compared prevailing algorithms for the FVC estimation of row crops from World-View 3. In our results, the physical model-based method and empirical method achieved satisfactory accuracy for the FVC of row crops compared to the field measurement. Different from the empirical method, the physical model-based method is automatically operated without any prior knowledge on the land cover, with no human interaction and empirically obtained parameters. This method can overcome the difficulties in determining the parameters in the empirical method. Therefore, the physical model-based method has

the potential to operationally estimate the FVC of row crops from a high spatial resolution image. To achieve good calculation accuracy, the choice of model in the physical model-based method is very important. In this study, the MFS + BPNN considered a row crops model (MFS model) to estimate the FVC of row crops, and thus the MFS + BPNN obtained an optimal accuracy (R = 0.9755, RMSE = 0.0312). MFS + BPNNs have the potential to estimate the FVC of row crops in precision agriculture, which indirectly predicts the early growth status of crops.

In addition, previous studies on FVC estimation mainly focused on kilometric resolution remote sensing data, a small amount of decametric resolution remote sensing data, and several kilometric resolution FVC products generated from SPOT-VET, MERIS, MODIS, and AVHRR sensors [9,25,26,44–49]. This study compared prevailing algorithms for FVC estimation of row crops from 2 m-resolution data (World-View 3), which is a new approach. Since the sensor can further detect the row structure of crops when the spatial resolution of the sensor is improved, the row crops model (MFS model) needs to be used to estimate the FVC of row crops. Figure 5b showed a density plot of the relationship between the NDVI and FVC estimated by the PROSAIL + BPNN. In this result, although the scatter points were exhibited an exponential relationship, estimated FVC values were discrete distribution. The main reason was that the row structure does not take into account the continuous crop model (PROSAIL model), which led to a mixed relationship between row closure reflectance and the reflectance of the between-row that cannot be calculated, and further affected the continuity of estimated FVC based on remote sensing reflectance data. Moreover, since the continuous crops did not have a between-row, the continuous crop model used the calculation method of the vegetation component to calculate the multiple scattering of this area, which led to an overestimation of reflectance in the canopy of row crops. FVC and leaf area index are closely related: leaf area index is the optical thickness of the vegetation. The optical thickness of the vegetation affects multiple scattering, eventually affecting the reflectance. Based on the above logic analysis, we suggested that the multiple scattering of the model influences FVC estimation. This conclusion is well shown in Figure 5b. In Figure 5b, when NDVI values are more than 0.8, the FVC of row crops calculated by the PROSAIL + BPNN was greater than the FVC of row crops calculated by the MFS + BPNN (red circle in Figure 5b). This illustrated that the vegetation structure is not negligible for FVC estimation of the heterogeneous canopy in high spatial resolution images.

In this study, we chose the early growth status of crops (i.e., row crops) as a target of FVC estimation. The constraint conditions in C-SAM (C-SMA-2 and C-SMA-5) were considered to eliminate the abnormal points of the estimated FVC (FVC > 1 or FVC < 0 in Figure 4c,e), but the accuracy of the estimation of the FVC of row crops was not particularly obvious. Similar to the previous study [21], the FVC estimated by SMA had the overestimation (Figure 5c–f) in the early growth status of crops, and was not suitable for FVC estimation in the early growth stage of crops. Compared to the estimation results of SMA, the PROSAIL + BPNN and the empirical method presented improved calculation accuracy. There is still a slight overestimation in these algorithms. Previous studies have found that the FVC is affected by soil background reflectance [21,50]. The soil background in the row crops is large, hence we also support this conclusion. Analyzing the algorithms involved in the comparison, MFS + BPNN was a nice choice to estimate the FVC of row crops, which also addressed the overestimation in the early growth status of crops.

NDVI was usually strongly related to FVC at a small regional scale and used in the empirical method and SMA [1,7,10]. Therefore, if the estimated FVC had a strong relationship with NDVI at a small regional scale, it could indicate that the estimated FVC was reasonable. In this study, the MFS + BPNN, PROSAIL + BPNN, U-SMA-2, C-SMA-2, U-SMA-5, C-SMA-5, and empirical method had a relationship with NDVI. Among them, strongly related with NDVI was the FVC estimated by the MFS + BPNN and empirical method, which further implied that these two algorithms were more reasonable for estimating the FVC of row crops. Different from the empirical method, the MFS + BPNN

was an automation algorithm, which can form a streamlined workflow to estimate the FVC of row crops from World-View 3. Therefore, the MFS + BPNN is more suitable for the generation of products in the high spatial resolution image. In our study, the FVC of row crops and NDVI have exponential relationships. The same relationship can be found in previous studies [30,31], which implies that the range of FVC of row crops is below the level at which NDVI saturates. We suggest that the FVC is more suitable as a greenness indicator to more accurately reflect farmland growth, and will have more application prospects in precise agriculture.

## 6. Conclusions

To find a reasonable algorithm for estimating the FVC of row crops in the high spatial resolution images, further serving precision agriculture, this study introduced a canopy reflectance model of row crops coupled with BPNNs to estimate the FVC of row crops. Comparing the prevailing algorithms, the accuracy for the canopy reflectance model of row crops coupled with BPNNs was higher, which again shows that the canopy reflectance model of row crops is more suitable for estimating the FVC of row crops. As high spatial resolution remote sensing images are increasingly used in precision agriculture, the vegetation structure of the heterogeneous canopy needs to be considered in the FVC estimation. In future FVC estimation, the reflectance model of the heterogeneous canopy will require more and more attention. Therefore, the MFS + BPNN has the potential to estimate the FVC of row crops based on a high spatial resolution image.

**Author Contributions:** X.M., L.L., J.D., F.Z. and B.H. conceived the idea. X.M. completed the implementation of the algorithm program, data analysis, and drafted the manuscript. All authors contributed to the editing of the manuscript. All authors have read and agreed to the published version of the manuscript.

**Funding:** This research was funded by Ph.D. starts funds in Xinjiang University, grant number 620321021.

**Institutional Review Board Statement:** Not applicable.

**Informed Consent Statement:** Not applicable.

**Acknowledgments:** The authors would like to thank the anonymous reviewers and the editor for the constructive comments and suggestions, all of which have led to great improvements in the presentation of this article. We also thank the help from Yong Liu (Lanzhou University) and Ying Liu (Xinjiang University).

**Conflicts of Interest:** The authors declare no conflict of interest.

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
