# Peer review of "Estimating Fractional Vegetation Cover of Row Crops from High Spatial Resolution Image"

_remotesensing, doi:10.3390/rs13193874_

Round 1
Reviewer 1 Report
The authors used three methods empirical method, Spectral Mixture Analysis and Physical model based method to estimate FVC of row crops from World view3 images. For physical based method, authors have used the modified physical reflectance model from the study (6). The description of these methods are good; however there is no novelty in the manuscript or has not been presented well.
Author Response
Reviewer #1
The authors used three methods empirical method, Spectral Mixture Analysis and Physical model based method to estimate FVC of row crops from World view3 images. For physical based method, authors have used the modified physical reflectance model from the study (6). The description of these methods are good; however there is no novelty in the manuscript or has not been presented well.
Response: Thanks for the reviewer’s insightful comment. Indeed, the part of the content needs further improvement to better present the advantages of the manuscript. In this revision, we have described the method (section 2.2.2, 2.2.3, and 2.3.2). Please refer to page 3, lines 118-135; page 4, lines 132-135, page 6, lines 210-226. Moreover, we analyzed the results in-depth and discussed them in the discussion section to show the advantages of the manuscript. Please refer to page 13, lines 371-381 lines 389-391; page 14, lines 402-407.

Reviewer 2 Report
This paper handles a significant field of study consisting in estimating the fractional vegetation cover (FVC) of row crops. However, some changes are recommended:
- Minor English checking is required.
- Using half of the training data to validate the model is too large. You can partition the total data as follow: 80% for training, 10% for validation, and 10% for testing.
- In machine learning, when we want to solve a regression task, we always use the mean square error (MSE) as a cost function because it is the best estimation of the model’s error and not the error summation.
- In line 18, there is no table 3, please, substitute Table 3 with Table2.
- In Table 2, could you explain why the same model features (inputs) are used as outputs? As I understand from your objective, you should use the three spectral bands as inputs and the FVC as output because your objective is to estimate the FVC.
- Could you justify the use of 6 hidden nodes in the hidden layer? I would prefer if you experiment with other numbers of hidden nodes and see what is the best configuration of your model.
- Please, try to organize the writing of the paper so as to remove the blank space on pages 8 and 9.
Author Response
Reviewer #2
This paper handles a significant field of study consisting in estimating the fractional vegetation cover (FVC) of row crops. However, some changes are recommended:
Minor English checking is required.
Response: Implemented. We had repeatedly checked full-text English and modified the minor mistake in English in the manuscript. Please refer to:
Page 2, lines 61-62, line 80, line 83, line 85, lines 94-96;
Page 3, lines 101;
Page 4, lines 138-140, lines152-155, line 159, line 164, lines 169-170;
Page 5, lines 175-176, lines 182-183, line 188, line 191, lines 193-194;
Page 7, line 231, line 243, line 248
Page 8, line 249, lines 255-257, line 260, line 262, line 266;
Page 9, line 298, lines 302;
Page 10, line 312;
Page 11, lines 323-324, line 326;
Page 12, line 335;
Page 13, line 349, lines 351-352, lines 353-354, line 356, line 363, lines 367-368, line 396;
Page 14, lines 401-403, line 418.
Using half of the training data to validate the model is too large. You can partition the total data as follow: 80% for training, 10% for validation, and 10% for testing.
Response: Implemented. We recalculate BPNNs in accordance with your recommendations, i.e., 80% for training, 10% for validation, and 10% for testing. Not only that, but we also chose a proportion of 6: 2: 2 to allocate training dataset, validating dataset, and testing dataset. In MFS+BPNNs, the FVC of row crops exists 0 when NDVI from 0-0.35, see Fig. 1(d) in this answer. In the statistics, we also found that RMSE in Fig. 1(e) is less than RMSE in Fig. 1(f) in this answer. Similarly, PROSAIL+BPNNs showed similar results in Fig. 2 in this answer. Therefore, the distribution of learning datasets is 6: 2: 2 more appropriate (Fig. 1-2 in the answer, RMSE=0.0305 for MFS+BPNNs and RMSE=0.0665 for PROSAIL+BPNNs). Due to the Figures 1-2 in this answer that cannot be present in this format, please refer to our attachments.
In machine learning, when we want to solve a regression task, we always use the mean square error (MSE) as a cost function because it is the best estimation of the model’s error and not the error summation.
Response: The mean square error (MSE= F=1/N *Summations[(y)i-(y^)i]^2) is a common cost function. Strictly speaking, the common cost function is a cost function similar to MSE. The equation is as shown below:
F=1/2 *Summations[(y)ij-(y^)ij]^2 |
(1) |
The cost function is a lot (see papers and doctoral thesis, the following [1-2]), no matter what cost function, its main purpose in mathematics is to find minimum deviation. In this study, we choose absolute error, and the equation is as shown below:
F=Summations[(y)ij-(y^)ij] |
(2) |
The MSE (Eq. (1)) is not perfect for this study data. Since we initialized our weights randomly with values close to 0, this expression will be very close to 0, which will make the partial derivative nearly vanish during the early stages of training. This article had been explained in detail: https://rohanvarma.me/Loss-Functions/. The figure below is the result of our test. Even if the data structure is double-precision floating-point (value domain is 0~2^64), the calculated derivative (see Eq. (8-9) in the modified manuscript) will generate invalid values. The main reason for this phenomenon is that there is a value out of range after the square, e.g., (4.5469983e+148)^2.
To address the problem of programming, we used absolute error as a cost function. The figure below is the result of our test. In addition, to address out of range, the cumulative error in the BPNNs model is more convenient to analyze, which is beneficial to find optimal parameters in BPNNs.
Due to the figures and equations that cannot be present in this format, please refer to our attachments.
References:
[1] Caicedo, R. and J. Pablo (2014). "Optimized and automated estimation of vegetation properties: Opportunities for Sentinel-2." Problems of Economic Transition 24(3): 86-98.
[2] Rivera, J. P., et al. (2013). "Multiple Cost Functions and Regularization Options for Improved Retrieval of Leaf Chlorophyll Content and LAI through Inversion of the PROSAIL Model." Remote Sensing 5(7): 3280-3304.
In line 18, there is no table 3, please, substitute Table 3 with Table2.
Response: What you said should be line 189 in the original manuscript, we have modified this typo. Please refer to page 6, line 217.
In Table 2, could you explain why the same model features (inputs) are used as outputs? As I understand from your objective, you should use the three spectral bands as inputs and the FVC as output because your objective is to estimate the FVC.
Response: Sorry. Because of care, we have a mistake, the input layer is the three spectral bands, and the output layer is FVC. The revision is shown in Table 2. Please refer to page 7, line 227.
Could you justify the use of 6 hidden nodes in the hidden layer? I would prefer if you experiment with other numbers of hidden nodes and see what is the best configuration of your model.
Response: We look for related papers of FVC inversion, try to find the a priori knowledge to set the number of hidden nodes. We refer to a paper published in remote sensing of environment (Fractional vegetation cover estimation algorithm for Chinese GF-1 wide field view data). In this paper, the hidden layer was set to 6. According to your advice, we tried other numbers of hidden nodes. Analyzing the variation of deviation during the network cycle (Quantitative analysis, Fig. 3 in this answer), we identified the best configuration of our model. The modification sees page 6, lines 210-226. Due to the figures and equations that cannot be present in this format, please refer to our attachments.
Please, try to organize the writing of the paper so as to remove the blank space on pages 8 and 9.
Response: Agreed. To address this problem, we have added a text description in section 2.2. Since then, the text of the manuscript increases, the relevant section have been moved, which makes up for the blank part on pages 8 and 9 in the original manuscript. Considering the typography in this place, we had moved this figure to the front of the text. The modification sees pages 9-10 in the revised manuscript.

Reviewer 3 Report
This is an interesting article, as the authors note, remote sensing is increasingly used in precision Agriculture, I think it contributes towards the accurate estimation of FVC of row crops from high-resolution images which can help in predicting field-scale crop yield.
The manuscript is well written and attractive, however, there are some areas need revision:
- Page 7, para 3.2.: Regarding the Worldview 3 image, the acquisition date and time need to be included. Maybe you can think of adding a table for the whole data used in this study with an explicit sentence (brief description)
- Page 7, para 3.2.: In addition to FLAASH , It is really useful to include what other software(s) did you use (free or commercial) for the estimation of FVC by the different algorithms , this needs an explicit sentence or two at end of paragraph.
- Page 8, para 3 : Figure 5, a typo, correct (b).
Author Response
This is an interesting article, as the authors note, remote sensing is increasingly used in precision Agriculture, I think it contributes towards the accurate estimation of FVC of row crops from high-resolution images which can help in predicting field-scale crop yield. The manuscript is well written and attractive, however, there are some areas need revision:
Page 7, para 3.2.: Regarding the Worldview 3 image, the acquisition date and time need to be included. Maybe you can think of adding a table for the whole data used in this study with an explicit sentence (brief description)
Response: Implemented. We had added the Table 3 in this revision and had simplified description. Please refer to page 9, lines 280-281, line 291.
Page 7, para 3.2.: In addition to FLAASH , It is really useful to include what other software(s) did you use (free or commercial) for the estimation of FVC by the different algorithms , this needs an explicit sentence or two at end of paragraph.
Response: Thanks for your advice. In this revision, we explained this unclear position. For the preprocessing of remote sensing data, we use ENVI software (commercial), including radiometric calibration, geometric correction, and atmospheric correction. Where atmospheric correction uses the FLAASH module in the ENVI software. These estimated algorithms of FVC had no ready-made software. We use computational code to calculate FVC. The code platform of computer language is IDL (8.3 version, Microsoft Windows, Win32 x86 m64). Please refer to page 9, lines 381-390.
Page 8, para 3 : Figure 5, a typo, correct (b).
Response: The typo you point out in Figure 5 on Page 8, para 3, is this Figure 3? Since the red area in the sub-graphs of Figure 3 is wolfberry, we use Figure 3 directly without adding the subscript. In addition, because the new figure was inserted in front of this figure, Figure 3 at this time is Figure 4 in the revised manuscript. Please refer to page 9, line 296.

Reviewer 4 Report
The article contributes to the Remote Sensing journal. However, it is important to add some comments that I already did to the article body.
In Figure 1 Please change the letter, and the Row direction too. For example
World-View 3 image in Zhongwei city (b) and World-View 3 image of farmaland in Zhongwei city (c).
In line 223 please describe the equipment that you used in NDVI measurements.
In Figure 2(d) write corn instead of wheat
In the results section please write homogeneous 4 (c-f) or 4(c-f) in all paragraphs
Also in the article body, write Figure 5 or Figure. 5 (with or without point)
In Figure 5 change b instead of d and also in the same figure change g instead of f

Author Response
The article contributes to the Remote Sensing journal. However, it is important to add some comments that I already did to the article body.
In Figure 1 Please change the letter, and the Row direction too. For example World-View 3 image in Zhongwei city (b) and World-View 3 image of farmland in Zhongwei city (c).
Response: Implemented. We corrected the letter and row direction in Figure 1, e.g., (b) World-View 3 image in Zhongwei city, and (c) World-View 3 image of farmland in Zhongwei city. Similarly, we had modified a typo of letters in the manuscript. The modification sees page 7, lines 337-341. One of the reviews emphasized the typographic issue in the manuscript; hence, we had moved this figure to the front of the text. In addition, because the new figure was inserted in front of this figure, Figure 1 at this time is Figure 2 in the revised manuscript. Due to the figures and equations that cannot be present in this format, please refer to our attachments.
In line 223 please describe the equipment that you used in NDVI measurements.
Response: Agreed. In this study, we use the SpectroSense 2+ multi-channel canopy spectrum measurement system (Due to the figures that cannot be present in this format, please refer to our attachments) to measure NDVI. This equipment can load the sensors for the red band and near-infrared band, thereby achieving the target of NDVI. To distinguish between the common canopy analyzer, we use its product name in the revision. The modification see page 8, line 260-261.
In Figure 2(d) write corn instead of wheat
Response: Implemented. We modified wheat to corn in Figure. 2(d). Similarly, we had modified the typo of letter in the manuscript. The modification see page 8, line 264; page 9, line298. Due to the figures that cannot be present in this format, please refer to our attachments.
In the results section please write homogeneous 4 (c-f) or 4(c-f) in all paragraphs
Also in the article body, write Figure 5 or Figure. 5 (with or without point)
Response: Agreed. We have unified the format of figure (without point). At this time, because the new figure was inserted in front of this figure, Figure 4 is Figure 5 and Figure 5 is Figure 6 in the revised manuscript. The modification sees page 9, line 303, page 11, line 326, line 328.
In Figure 5 change b instead of d and also in the same figure change g instead of f
Response: Implemented. We have modified original Figure 5(d) to Figure 5(b). Similar, Figure 5(f) has also been modified to Figure 5(g). Because the new figure was inserted in front of this figure, Figure 5 at this time is Figure 6 in the revised manuscript. The modification sees page 12, line 340; page 11, line 332. Due to the figures and equations that cannot be present in this format, please refer to our attachments.

Round 2
Reviewer 1 Report
The manuscript has been revised very well.
The prevailing algorithms are now explained in detail.
The authors have attempted to use the canopy reflectance model of row crops coupling backward propagation neural networks (BPNNs) to estimate the FVC of row crops. And now the algorithm has been explained nicely.
Authors can reconsider the first line of conclusion. It sounds incomplete.
The paper is suitable for publication.